# The Sec7 N-terminal regulatory domains facilitate membrane-proximal activation of the Arf1 GTPase

Brian C Richardson[1,2], Steve L Halaby[1,2], Margaret A Gustafson[1,2], J Christopher Fromme[1,2]*

[1]Department of Molecular Biology and Genetics, Cornell University, Ithaca, United States; [2]Weill Institute for Cell and Molecular Biology, Cornell University, Ithaca, United States

**Abstract** The Golgi complex is the central sorting compartment of eukaryotic cells. Arf guanine nucleotide exchange factors (Arf-GEFs) regulate virtually all traffic through the Golgi by activating Arf GTPase trafficking pathways. The Golgi Arf-GEFs contain multiple autoregulatory domains, but the precise mechanisms underlying their function remain largely undefined. We report a crystal structure revealing that the N-terminal DCB and HUS regulatory domains of the Arf-GEF Sec7 form a single structural unit. We demonstrate that the established role of the N-terminal region in dimerization is not conserved; instead, a C-terminal autoinhibitory domain is responsible for dimerization of Sec7. We find that the DCB/HUS domain amplifies the ability of Sec7 to activate Arf1 on the membrane surface by facilitating membrane insertion of the Arf1 amphipathic helix. This enhancing function of the Sec7 N-terminal domains is consistent with the high rate of Arf1-dependent trafficking to the plasma membrane necessary for maximal cell growth.

*For correspondence: jcf14@cornell.edu

**Competing interests:** The authors declare that no competing interests exist.

## Introduction

In eukaryotes, trafficking of membranes and membrane proteins from their site of synthesis at the endoplasmic reticulum through the secretory pathway proceeds via the Golgi apparatus. This trafficking, mediated by membrane vesicles and tubules, must achieve the specificity to ensure that proteins are targeted to the correct destinations and with the correct post-translational modifications while retaining the throughput required to provide the bulk of the lipid content of the plasma membrane's growth over the course of cell division. Much of this process is controlled at the *trans*-Golgi network (TGN), where the final processed forms of proteins are recognized by cargo adaptors and packaged into vesicles for transport to endosomes, lysosomes, and the plasma membrane (*Guo et al., 2014*).

In both humans and model organisms, Arf GTPases act as the central coordination point for vesicle formation at the TGN, directly regulating a cascade of cargo adaptors (*Donaldson and Jackson, 2011*). Arf activation is regulated in turn by guanine nucleotide exchange factors (GEFs) that exchange bound GDP for GTP (*Casanova, 2007*). The accompanying conformational change in the Arf GTPase rearranges 'switch' regions for recognition by effectors and exposes a myristoylated amphipathic N-terminal helix that inserts into the membrane (*Antonny et al., 1997*; *Goldberg, 1998*). Arf-GEFs thereby stand upstream of Arf GTPases as the initiator of the cascade of events leading to vesiculation of TGN cisternae.

At the TGN, the Arf1-5 GTPases are activated by the BIG1/2 proteins in humans (*Mansour et al., 1999*; *Yamaji et al., 2000*) and the Arf1/2 GTPases are activated by Sec7 in the model organism *Saccharomyces cerevisiae* (*Achstetter et al., 1988*; *Peyroche et al., 1996*). These TGN-localized

**eLife digest** The cells of plants, animals and other eukaryotes are subdivided into different membrane-bound compartments. One of these compartments – called the Golgi complex – has been likened to the 'Grand Central Station' of the eukaryotic cell, because it serves as the cell's centralized sorting compartment. Small, spherical structures called vesicles arrive at the Golgi complex from other cellular compartments, and the material within these vesicles is then sorted, packaged into new vesicles, and sent out to different destinations.

Regulatory proteins are responsible for making decisions about when to turn on different incoming and outgoing pathways to or from the Golgi complex. In particular, one regulatory protein named Sec7 controls many of the outgoing vesicles that leave the Golgi complex. Sec7 is a fairly large protein and has different parts, or domains, that regulate how the protein works. It was known that two of these regulatory domains (named 'DCB' and 'HUS') were required for Sec7 to work, but it was not known what these domains actually did.

Richardson et al. have now used a technique called X-ray crystallography to reveal the three-dimensional structure of the regulatory DCB and HUS domains of Sec7 from a species of yeast. The Sec7 protein has been conserved throughout evolution with few changes, and so the structure of this yeast protein is expected to be the same as that of the corresponding protein in humans. Unexpectedly, Richardson et al. discovered that DCB and HUS are not structurally separate domains and actually form a single 'DCB/HUS' domain.

Biochemical experiments then showed that the DCB/HUS domain helps Sec7 work on the surface of membranes. One of the jobs of Sec7 is to insert another regulatory protein (called Arf1) into the membranes of the Golgi complex, and these new findings reveal that the DCB/HUS domain helps Sec7 overcome the challenges associated with this task.

Researchers have now uncovered structural information for approximately half of the Sec7 protein, and so an important future challenge will be to reveal the structure of the remaining regulatory domains of Sec7. This achievement will help researchers to figure out how all of the parts of Sec7 work together to control how this protein makes decisions.

Arf-GEFs are universally conserved in eukaryotes, and their importance is underscored by neurological disorders associated with mutations in the *BIG2/ARFGEF2* gene (*Banne et al., 2013*; *Sheen et al., 2004*).

The TGN-localized Arf-GEFs comprise roughly 1800 functional residues including a central catalytic domain of ~200 residues mediating nucleotide exchange (*Chardin et al., 1996*; *Morinaga et al., 1996*; *Peyroche et al., 1996*). This catalytic domain, commonly called the Sec7 domain, will be referred to here as the GEF domain for clarity. Conserved domains C-terminal to the GEF domain, termed HDS1-4 (homology downstream of Sec7) (*Mouratou et al., 2005*), act to integrate the signals from several small GTPases, including Arf1 itself, to switch Sec7 from a strongly autoinhibited to a strongly autoactivated form (*McDonald and Fromme, 2014*; *Richardson et al., 2012*; *Richardson and Fromme, 2012*). The N-terminal DCB (dimerization and cyclophilin binding) and HUS (homology upstream of Sec7) domains, though conserved in all Golgi Arf-GEFs and essential for Golgi function (*Ramaen et al., 2007*), remain more of an enigma. Numerous functions have been proposed, including cyclophilin binding, regulated dimerization, Arl1 and Arf1 GTPase binding, and enhancement of GEF activity, but studies are frequently contradictory and fail to assemble into a unified understanding of function (*Christis and Munro, 2012*; *Grebe et al., 2000*; *Lowery et al., 2013*; *Mouratou et al., 2005*; *Ramaen et al., 2007*; *Richardson et al., 2012*).

Here, we present the crystal structure of the regulatory DCB and HUS domains from *Thielavia terrestris* Sec7 and demonstrate that they form a single continuous structural domain. We find that dimerization of this N-terminal domain is not conserved and that Sec7 dimerization is primarily mediated by the C-terminal HDS4 domain. We describe a new function of the DCB/HUS regulatory domain: this domain enhances the activation of Arf1 in a manner dependent on both lipids and the N-terminal membrane insertion element of Arf1, implying a role in chaperoning the insertion of Arf1 into the Golgi membrane.

## Results

### The DCB and HUS regions form a unified structural domain

To gain structural insights into the function of the DCB and HUS domains, we purified and attempted to crystallize constructs from ten species, including *S. cerevisiae* and the thermophiles *Chaetomium thermophilum, Myceliophthora thermophila*, and *T. terrestris* (*Amlacher et al., 2011*; *Berka et al., 2011*). Following correction of the annotated intron (*Supplementary file 1*), a construct comprising residues 1–458 of *T. terrestris* produced crystals diffracting to 2.6 Å. Using experimental phase determination by single-wavelength anomalous diffraction of selenomethionine-substituted protein, the structure of the DCB and HUS domains was determined with four copies in the asymmetric unit. The best resolved chains in the model lack only unresolved loops and the final ~35 conserved residues of the HUS domain absent from the crystallized construct (*Figure 1A* and *Figure 1—figure supplement 1*, *Table 1*). Extensive crystal contacts between chains A and B stabilize them sufficiently to produce excellent electron density; the N-terminal portions of chains C and D, with fewer crystal contacts, were less well resolved and had only enough visible density to be modeled as copies of chains A and B (*Figure 1—figure supplements 1* and *2*).

The DCB and HUS domains together form a single continuous armadillo repeat joined by a conserved ~1400 Å$^2$ hydrophobic interface, comprising helices 7 and 8 from the DCB domain and helices 10 and 11 from the HUS domain. (*Figure 1A* and *Figure 1—figure supplement 3*). We therefore regard the N-terminal regulatory region of Sec7 as a single DCB/HUS structural domain. In most species, a poorly conserved ~100 amino acid linker connects the conserved DCB and HUS domains. In the *T. terrestris* DCB/HUS domain structure, residues 245–271 of this interdomain region (between helices 8 and 10, including helix 9) are resolved in their packing against the armadillo repeat, and the remaining 46 and 21 residue stretches are unresolved in the structure. Interestingly, the mutation E209K in human BIG2 associated with neuronal disease (*Sheen et al., 2004*) maps to this unresolved region. A striking conserved positively charged patch is seen at the interface of helices 7 and 10 (*Figure 1B*).

### Sec7 dimerizes via the HDS4 domain

The crystallized construct forms a dimer in the crystal via its C-terminal helices 14 and 15, with helix 15 of each copy aligned antiparallel to each other. Dimerization is known to occur in N-terminal constructs of Sec7 and its homologs (*Grebe et al., 2000*; *Ramaen et al., 2007*; *Richardson et al., 2012*), and small-angle X-ray scattering (SAXS) analysis confirmed that the observed dimerization mode occurred in solution as well as in the crystal (*Figure 2A*). However, the fact that the construct dimerizes via the end of a DCB/HUS domain construct that was truncated raised concern, as truncation of helical domains has been shown in the past to result in artificial dimer interfaces (*Richardson et al., 2009*). Therefore, we purified a longer *T. terrestris* Sec7 construct comprising the entire DCB/HUS domain (residues 1–492), for analysis by SAXS. While slight divergence is seen, the intact *T. terrestris* DCB/HUS domain SAXS data closely fits that predicted from a monomer (*Figure 2B*). This strongly suggests that the observed dimerization interface is an artifact resulting from truncation of the C-terminal helix of the HUS domain, although it is possible that this dimerization mode represents a regulated conformation of the N-terminus.

These observations, together with a report that a portion of the HDS4 domain mediates dimerization (*Mariño-Ramírez and Hu, 2002*), prompted us to perform an in-depth characterization of the multimerization behavior of different Sec7 constructs from both *T. terrestris* and *S. cerevisiae*. Many constructs failed to yield reliable SAXS data, so we turned to multi-angle light-scattering (MALS) to identify the oligomeric state of a series of truncation constructs. We found that full-length *T. terrestris* Sec7 and the corresponding *S. cerevisiae* Sec7$_f$ (residues 203–2009) construct were both dimeric (*Figure 2C*). Strikingly, removal of the C-terminal HDS4 domain resulted in a monomer for constructs from both species, whereas a construct corresponding to a predicted coiled-coil from the HDS4 domain (*Mariño-Ramírez and Hu, 2002*) was itself dimeric. This indicates that the HDS4 domain mediates dimerization of the intact Sec7 proteins from both *T. terrestris* and *S. cerevisiae*.

To examine the role of the HDS4 domain in cells, we attempted to introduce a GFP-Sec7ΔHDS4 construct into cells lacking the essential *SEC7* gene. Virtually all such cells were unable to grow, similar to the previously described lethality arising from removal of the HDS2-4 domains (*Figure 3A*)

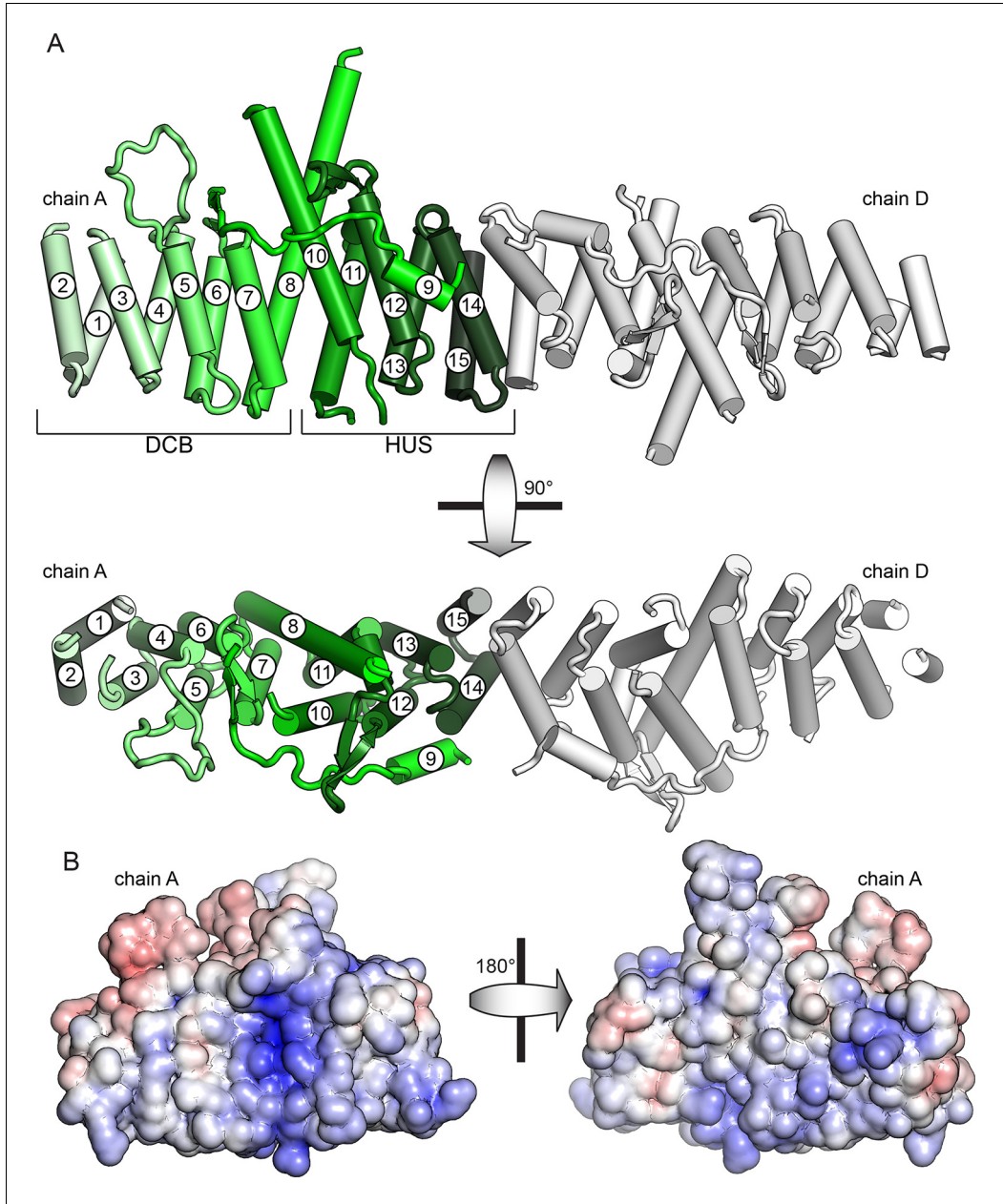

**Figure 1.** Crystal structure of *T. terrestris* Sec7 DCB/HUS domain (residues 1–458) (**A**) Chains A (green) and D (white) are shown; chain A is colored light to dark N to C, and helices are numbered N to C. The entire asymmetric unit is shown in supplement 1. Electron density is shown in supplement 2. The DCB/HUS interface is shown in supplement 3. (**B**) The charge potential surface of chain A alone, as calculated by APBS (*Baker et al., 2001*; *Dolinsky et al., 2007*), is colored on a red-white-blue gradient from -10 kT/e to +10 kT/e.

The following figure supplements are available for figure 1:

**Figure supplement 1.** Asymmetric unit of crystal structure.

**Figure supplement 2.** Electron density of crystal structure.

**Figure supplement 3.** Magnified view of the DCB/HUS interface.

**Table 1.** Data collection and refinement statistics

|  | T. terrestris Sec7 DCB/HUS domain (residues 1-458) |
| --- | --- |
| Wavelength (Å) | 0.987 |
| Resolution range (Å) | 50 - 2.6 (2.64–2.6) |
| Space group | P 21 21 21 |
| Unit cell | a=62.472Å b=132.024Å c=247.664Å α=β=γ=90° |
| Total reflections | 569136 |
| Unique reflections | 59606 |
| Multiplicity | 9.5 (4.7) |
| Completeness (%) | 98.77 (88.46) |
| Mean I/sigma(I) | 8.46 (1.48) |
| Wilson B-factor | 65.26 |
| R-work | 0.2119 (0.3113) |
| R-free | 0.2568 (0.3665) |
| Number of atoms | 10944 |
| Macromolecules | 10887 |
| Water | 57 |
| Protein residues | 1383 |
| RMS(bonds) | 0.009 |
| RMS(angles) | 1.23 |
| Ramachandran favored (%) | 98 |
| Ramachandran outliers (%) | 0.075 |
| Clashscore | 7.63 |
| Average B-factor | 89.7 |
| Macromolecules | 89.8 |
| Solvent | 60.3 |

(*Richardson et al., 2012*). To examine whether this effect was due to mis-localization of the construct, we imaged an otherwise wild-type strain harboring the GFP-Sec7ΔHDS4 construct. Although we detected a low level of cytoplasmic mislocalization, GFP-Sec7ΔHDS4 was largely localized correctly to the TGN, as observed by its colocalization with endogenous Sec7-RFP (*Figure 3B*). In contrast, localization of the shorter GFP-Sec7ΔC+HDS1 construct was more severely compromised, as seen previously (*Richardson et al., 2012*). Therefore, the HDS4 domain is critically important for Sec7 function in cells but does not appear to play a major role in its localization.

Further C-terminal truncation of *S. cerevisiae* Sec7 resulted in a surprising series of observations. Whereas a construct lacking the HDS4 domain is monomeric, a construct lacking the HDS2-4 domains was an equilibrium mixture of monomeric and dimeric species (*Figure 2C*). Removal of additional domains, generating constructs lacking the HDS1-4 domains with or without the GEF domain, resulted in stable dimers. However, the corresponding constructs of *T. terrestris* Sec7 were monomeric (*Figure 2C*).

The simplest model accounting for dimerization of the isolated *S. cerevisiae* N-terminus, dimerization via the HDS4 domain, and loss of dimerization when the HDS1 domain is present, invokes an interaction between the HDS1 and DCB/HUS domains (*Figure 2D*). When present, the HDS1 domain likely masks a site by which the DCB/HUS domain dimerizes in its absence, but Sec7 retains its dimerization via the HDS4 domain. Whether this masked dimerization is an artifact or serves a regulatory purpose remains unknown, but the presence of an exposed interaction domain in truncated constructs provides an attractive explanation for the complicated oligomerization results seen in prior work (*Grebe et al., 2000*; *Ramaen et al., 2007*), as well as for the seemingly absolute

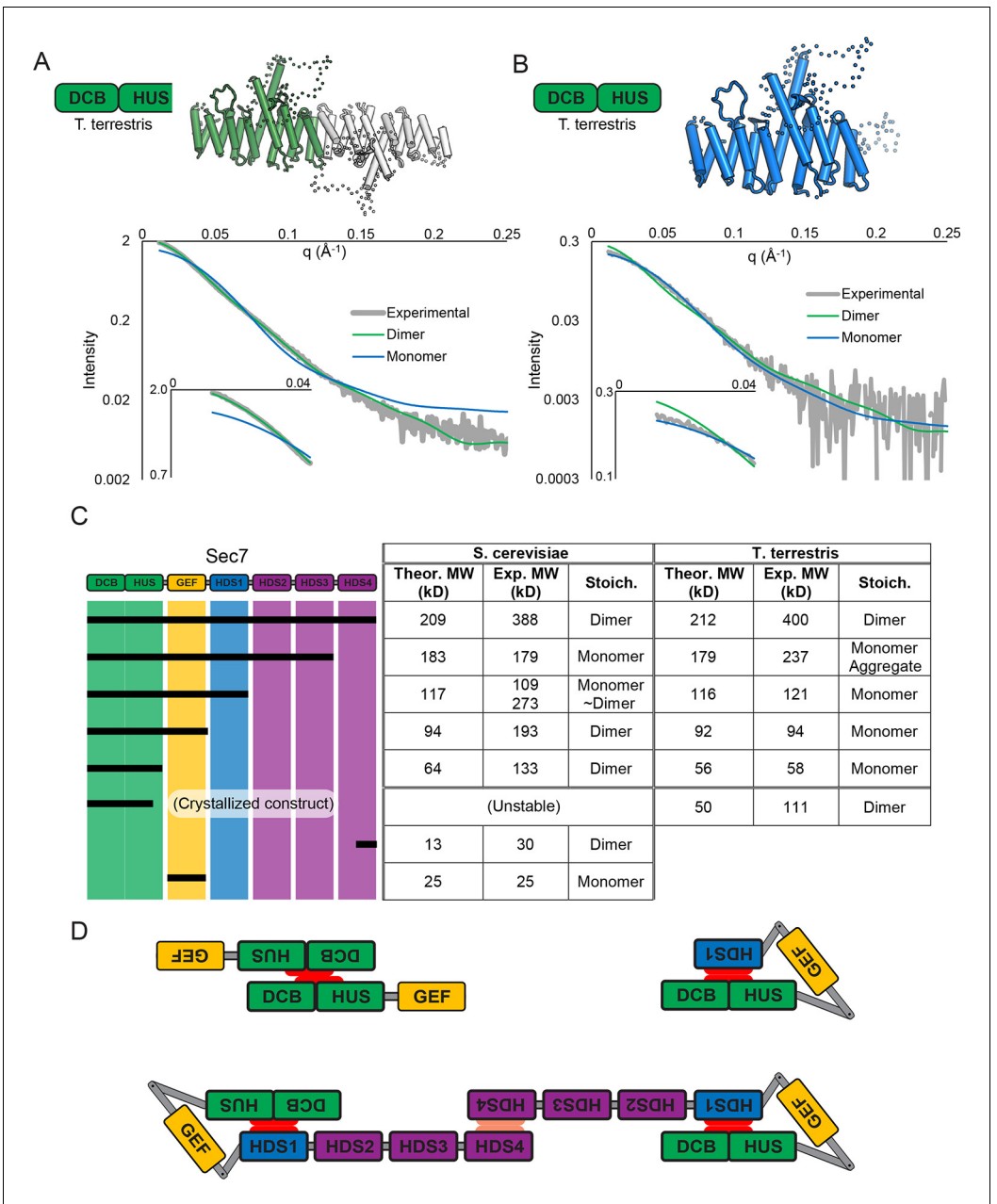

**Figure 2.** Sec7 dimerizes primarily via the HDS 4 domain. (**A**) CORAL (**Petoukhov et al., 2012**) was used to fit the *T. terrestris* Sec7 DCB/HUS domain structure (residues 1–458) to SAXS data collected on the same construct, accounting for the presence of unresolved loops modelled as dotted lines. For comparison, a similar calculation using only a single chain is shown with a significantly worse fit. (**B**) A single monomer from (**A**), fixing the previously modeled loops in place and adding additional residues at the C-terminus, was fit to SAXS data collected on *T. terrestris* Sec7(1–492); as the added region is expected to comprise an alpha helix in addition to a connecting loop, BUNCH was used in place of CORAL for modeling. A similar fit of the dimeric form is shown for comparison. (**C**) The solution molecular weights of the indicated *S. cerevisiae* and *T. terrestris* constructs were determined by MALS. Comparison to the predicted monomeric mass based on primary sequence yields the stoichiometry. SAXS results from *S. cerevisiae* Sec7ΔC are shown in supplement 1. Results from corresponding *S. cerevisiae* Gea2 constructs are shown in supplement 2. Original MALS traces of all *S. cerevisiae* and *T. terrestris* constructs are shown in supplement 3. (**D**) Schematic model of *S. cerevisiae* cis and trans interactions of truncated constructs. Red represents a hypothesized interface between HDS1 and DCB/HUS, the latter half of which can self-stabilize by dimerization in the absence of HDS1. Orange represents the HDS4 dimerization interface. *T. terrestris*

requirement of the DCB/HUS domain for soluble expression of most constructs containing HDS domains. Unfortunately, this latter phenomenon also precludes more thorough biochemical testing of this hypothesis, leaving this model partially speculative. It is also evident from the behavior of the *T. terrestris* constructs that dimerization of the DCB/HUS domain is not a conserved behavior. We note that this analysis does not exclude the possibility of membrane- or temperature-dependent effects on dimerization or possible influences on Sec7 dimerization by other proteins.

While reliable SAXS data could not be obtained from *T. terrestris* constructs including the GEF domain, modeling our N-terminal structure and a previously solved GEF domain structure (*Renault et al., 2002*) to SAXS data collected on *S. cerevisiae* Sec7(203–1017) (Sec7ΔC) suggests that the GEF domain is flexibly connected to the DCB/HUS domain rather than continuing the armadillo repeat (*Figure 2—figure supplement 1*). As such, it appears unlikely that the intact DCB/HUS construct (1–492) is prone to the same artifactual dimerization issues at its C-terminus as is the truncated Sec7(1–458) construct.

Given our finding that the HDS4 domain appears to be the primary mediator of Sec7 dimerization, we decided to investigate the multimerization state of the distantly related early-Golgi Arf-GEF Gea2 from *S. cerevisiae*, which has evolved without an HDS4 domain. MALS analysis revealed that Gea2 full-length and C-terminal truncation constructs are dimeric (*Figure 2—figure supplement 2*). This suggests that dimerization in some form is critical to the regulation of all the Golgi Arf-GEFs, and that in the absence of an HDS4 domain, N-terminal dimerization may have evolved to acquire additional importance.

## Sec7 DCB/HUS domain activity is lipid-dependent

We previously observed that the DCB/HUS domain confers a significant enhancement of activity on the GEF domain, in the physiologically relevant context of a myristoylated Arf1 substrate in the presence of liposome membranes (*Richardson et al., 2012*). We sought to investigate the mechanism for this enhancement on the basis of the DCB/HUS crystal structure. We used an established in vitro GEF activity assay taking advantage of the innate change in tryptophan fluorescence of Arf1 as it transitions from binding GDP to binding GTP (*Bigay and Antonny, 2005*; *Kahn and Gilman, 1986*) to track reaction kinetics without the need for artificial substrates (*Figure 4A*). Extending our previous results, we were surprised to observe that when we removed membranes from the reaction and truncated the N-terminal membrane insertion helix of Arf1 to preserve its ability to exchange nucleotide (ΔN17Arf1), the DCB/HUS domain of *S. cerevisiae* Sec7 no longer stimulated GEF activity. Indeed, the Sec7ΔC construct was slightly less active than the GEF domain alone, possibly due to a reduced rate of diffusion (*Figure 4A,B*). Therefore, the DCB/HUS domain stimulates GEF domain activation of Arf1 on membranes.

To determine whether the membrane-localized activation of Arf1 monitored by tryptophan fluorescence corresponded to the concomitant membrane association of activated Arf1 with membranes, we performed liposome flotation following a GEF-mediated exchange timecourse to measure the stable membrane association of activated Arf1 (*Figure 4—figure supplement 1*). The kinetics of Arf1 membrane insertion as measured in this assay directly correlated with the kinetics of Arf1 activation as measured by tryptophan fluorescence and further confirmed the role of the DCB/HUS domain in stimulating Arf1 activation at the membrane surface.

Similar results were also observed using equivalent *T. terrestris* constructs on the basis of tryptophan fluorescence (*Figure 4—figure supplement 2*). As the stimulatory effect is observed in both dimeric (*S. cerevisiae*) and monomeric (*T. terrestris*) constructs, we infer that DCB/HUS domain stimulation of GEF activity occurs independently of dimerization. Two potential mechanisms of action thereby present themselves: either the DCB/HUS domain binds to the membrane surface to recruit the GEF domain to its site of action, or the DCB/HUS domain assists in membrane-insertion of the N-terminal amphipathic helix of Arf1.

To distinguish between these two possibilities, we changed the lipids in the reaction mixture from physiologically relevant TGN-like liposome membranes to non-physiological DMPC/cholate micelles known to support nucleotide exchange of myristoylated Arf1 (*Kahn and Gilman, 1986*). In the presence of these micelles, the activity of both constructs on myristoylated Arf1 was higher, indicating an increase in the intrinsic rate of nucleotide exchange, yet the Sec7ΔC construct was again more active than the isolated GEF domain (*Figure 4C*). As this effect is not seen on ΔN17Arf1 in the presence of micelles, which displays a uniformly reduced rate of exchange, it cannot be attributed merely

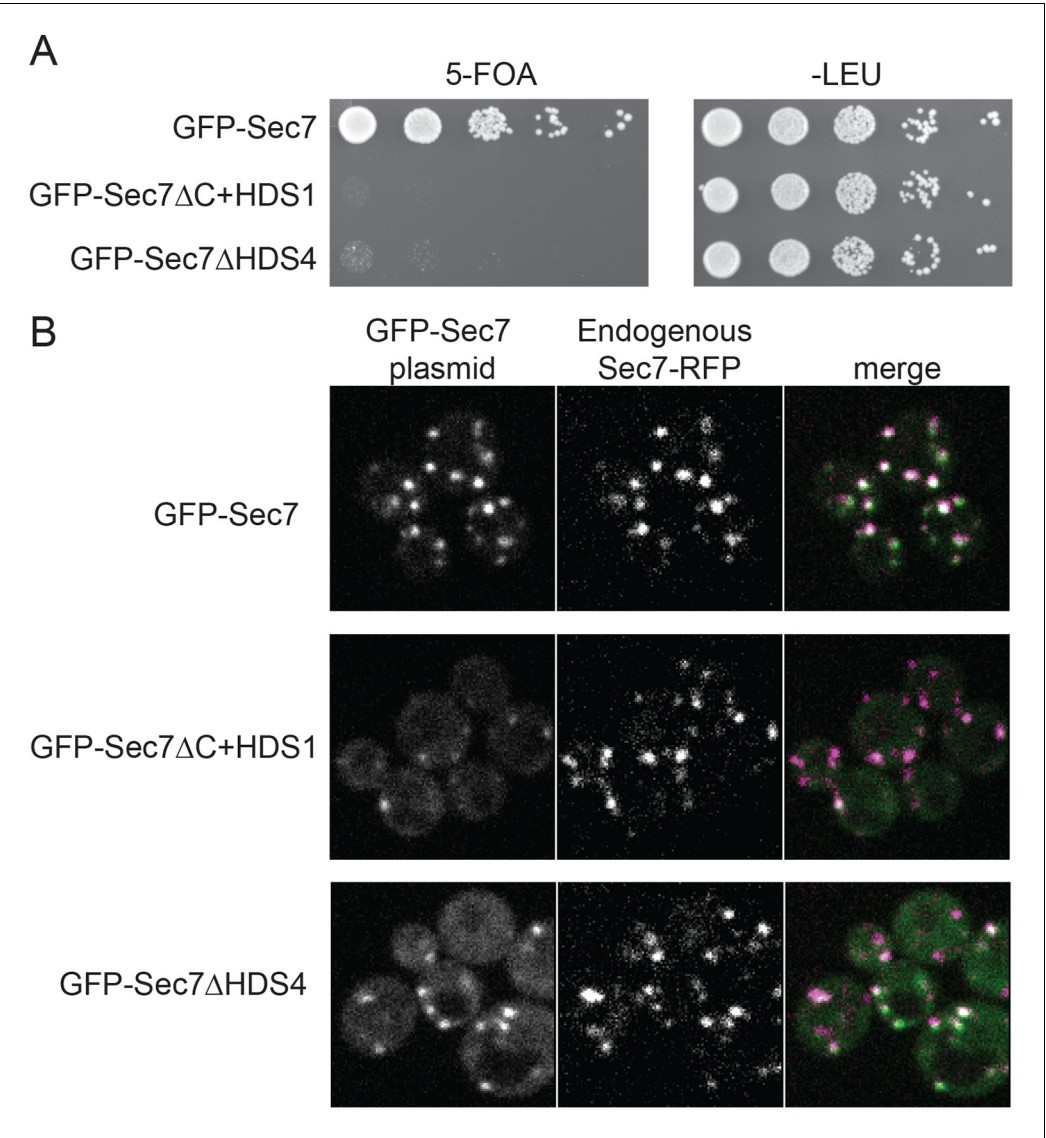

**Figure 3.** The HDS4 domain of Sec7 is important for function but dispensable for TGN localization. (**A**) Centromeric plasmids encoding GFP-Sec7 constructs expressed from the *SEC7* promoter were introduced into a *SEC7* plasmid shuffling strain (CFY409). Growth on 5-FOA measures the ability of the construct to complement the *sec7Δ* mutation. (**B**) The same plasmids were imaged in an otherwise wild-type strain expressing endogenously tagged Sec7-RFP[Mars] ('Sec7-RFP') (CFY578).

to the lipids themselves. In a separate experiment we introduced the L8K mutation into the membrane insertion helix of Arf1, which reduces its hydrophobicity and enables Arf1 activation in the absence of membranes (*Luo et al., 2007*; *Yoon et al., 2004*). We observed that the DCB/HUS domain did not stimulate activity of the GEF domain towards L8K-Arf1 in either the absence or presence of membranes (*Figure 4D*). Taken together, these results suggest that GEF stimulation by the DCB/HUS domain involves insertion of the Arf1 amphipathic helix into a hydrophobic and presumably lipidic environment, yet this effect is indifferent to the precise nature of the lipids.

Our interpretation of these observations is that the DCB/HUS domain assists in overcoming a kinetic activation barrier associated with insertion of the myristoylated Arf1 N-terminal helix into membranes. An alternative possibility, that the DCB/HUS domain has indiscriminate affinity for lipid surfaces and therefore increases the rate of membrane-localized encounters between the GEF domain and its substrate, is addressed further below.

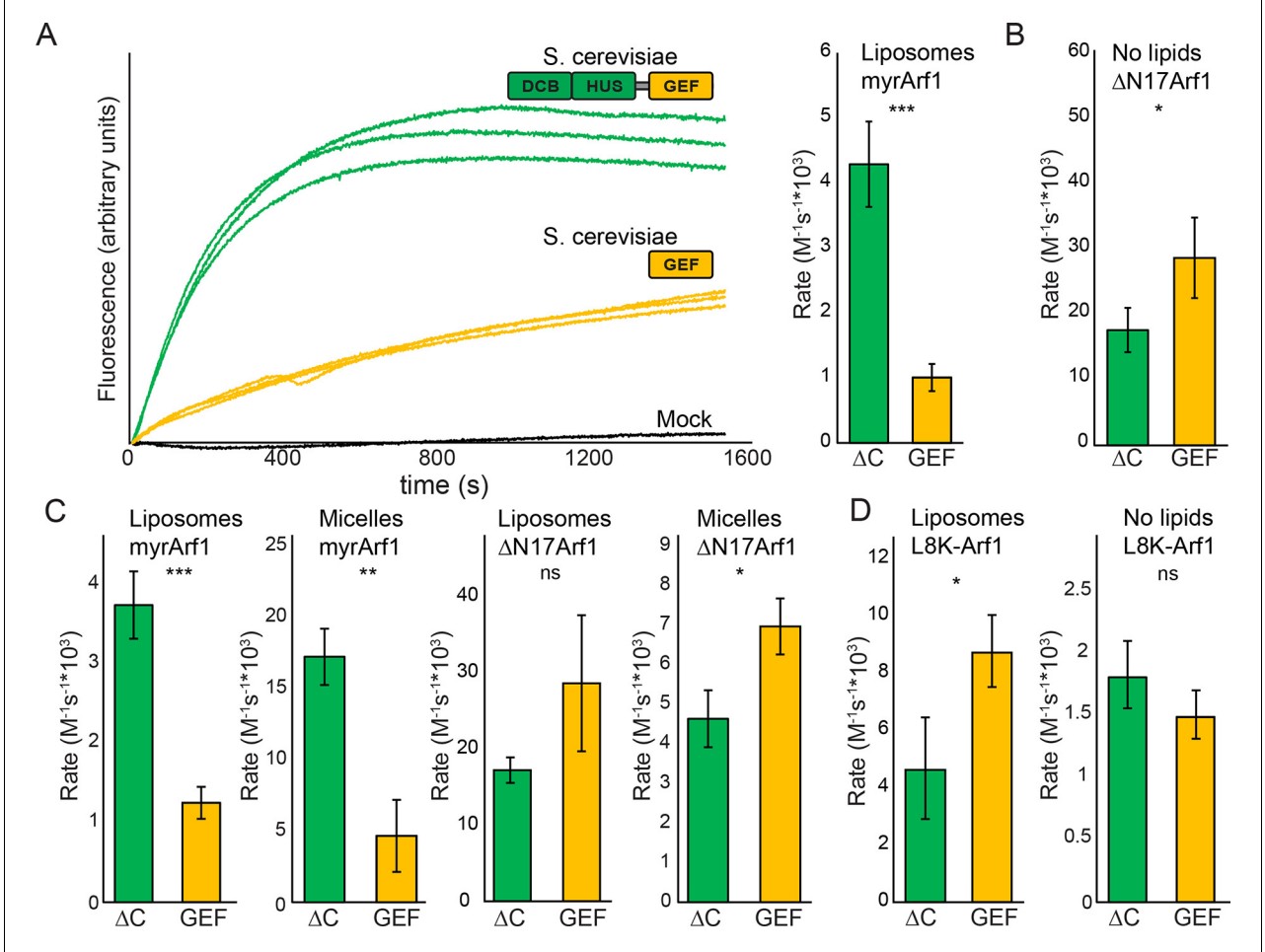

**Figure 4.** Stimulation of GEF activity by the DCB/HUS domain depends on the presence of lipids. (A) Triplicate nucleotide exchange curves of *S. cerevisiae* Sec7ΔC (green traces) or isolated GEF domain constructs (yellow traces) acting on myristoylated Arf1 substrate in the presence of liposomes are shown against a mock exchange reaction (black trace) (left). The curves were fit to a single exponential and normalized to Sec7 concentration to extract the exchange reaction rates (right). Error bars denote 95% confidence intervals, n=3. Purity of all constructs is demonstrated in supplement 1. These measurements of exchange by tryptophan fluorescence are compared to complementary measurements of Arf1 membrane binding in supplement 2. (B) Exchange reaction rates of *S. cerevisiae* Sec7 constructs acting on the ΔN17Arf1 substrate in the absence of lipids. GEF activity of *T. terrestris* constructs is shown in supplement 3. GEF activity following Arl1 preincubation is shown in supplement 6, with corresponding physical interaction analysis in supplements 4 and 5. (C) Parallel reactions of *S. cerevisiae* Sec7 constructs acting on myristoylated Arf1 and ΔN17Arf1 in the presence of liposomes and micelles. (D) Nucleotide exchange by *S. cerevisiae* Sec7 constructs on non-myristoylated L8K-Arf1 in the presence and absence of liposomes.

The following figure supplements are available for figure 4:

**Figure supplement 1.** Measurement of Sec7 GEF kinetics by liposome flotation.

**Figure supplement 2.** GEF activity of *T. terrestris* Sec7.

**Figure supplement 3.** GST pulldown analysis of Arf1 and Arl1 interaction with Sec7ΔC.

**Figure supplement 4.** Liposome pelleting analysis of Arf1 and Arl1 interaction with Sec7ΔC.

**Figure supplement 5.** Effect of Arl1 preincubation on Sec7 GEF activity.

**Figure supplement 6.** Purity of constructs used for kinetic assays.

## The surfaces of helices 4 and 6 mediate DCB/HUS activity

To pair these observations to the structure of the DCB/HUS domain, we performed a targeted functional screen by generating an extensive panel of mutants in *S. cerevisiae*, selecting conserved residues found at the surface of the *T. terrestris* DCB/HUS domain structure (*Figure 5A*). We investigated whether alanine or aspartate substitutions in these conserved surface residues resulted in cellular viability phenotypes at elevated temperature in a sensitized strain in which the levels of Arf1/2 have been knocked down by 90% through disruption of the *ARF1* gene (*Stearns et al., 1990*) (*Figure 5B*). The established *sec7-1* temperature sensitive mutation (S402L in *S. cerevisiae*) (*McDonold and Fromme, 2014*; *Novick et al., 1980*) maps to a conserved serine at residue 156 in the α7–8 loop (*Figure 6—figure supplement 1*). The leucine substitution arising from the *sec7-1* mutation likely perturbs the local structure of this region critical to forming the DCB/HUS interface. Several other mutants displayed temperature-sensitive growth phenotypes; the most significant growth defects were caused by mutations in residues located on a conserved surface of helices 4, 6, and 8, on the opposite side of the protein from the conserved positively charged patch (*Figure 5A,B*).

To explore whether any of these mutations affected localization of Sec7 to the TGN, we imaged mutant GFP-Sec7 plasmids in the sensitized *arf1Δ* strain. All of the new mutants that we examined expressed well and exhibited largely punctate localization (*Figure 6A*). Although some of the mutants were partially mislocalized to the cytoplasm, none were as severe as either the *sec7-1* mutant or previously identified mutants in the HDS1 domain (*Richardson et al., 2012*). For example, although the D297A/K301D/F305D mutant exhibited significant cytoplasmic mislocalization, it was also strongly localized to punctate structures (*Figure 6A*). In contrast, the *sec7-1* mutant only occasionally exhibits robust punctate localization. Therefore, the growth defects displayed by the new DCB/HUS domain mutants do not appear to be attributable to loss of proper localization.

To test whether any of the mutations affecting growth also affected GEF activity, we prepared mutant Sec7ΔC proteins for in vitro analysis. The use of the Sec7ΔC truncation enabled us to directly monitor the role of the DCB/HUS domain without confounding affects from the C-terminal HDS1-4 domains. While several mutants could not be purified, including *sec7-1*, a sufficient number representing both conserved regions were purified to allow us to test the involvement of each region in the DCB/HUS stimulatory activity (*Figure 6B,C*). We found that mutations on the surfaces of helices 4 and 6 (*S. cerevisiae* residues 297–305 and 364–368) reduced both cell viability and membrane-dependent GEF activity of Sec7ΔC, whereas mutations in the positively charged patch (i.e., *S. cerevisiae* residues Arg505 and Lys513) had little effect in vitro despite modestly affecting growth. None of the tested mutations reduced GEF activity in the absence of membranes (*Figure 6—figure supplement 2*), indicating the mutations that reduce membrane-proximal GEF activity do not simply cause misfolding of these contructs. Therefore, the surfaces of helices 4 and 6 are important for both GEF activity and cell viability.

A notable absence from both the structure presented here and the available structures of the GEF domain is the highly conserved 'HUS-box' located between helix 15 and the helix 16 truncated from our crystallized construct (*Mouratou et al., 2005*). A previous study reported that the HUS-box was required for a yeast 2-hybrid interaction between the DCB and HUS domains (*Ramaen et al., 2007*), but our structural data do not support such a role. To assess whether the HUS-box plays a role in GEF stimulation by the DCB/HUS domain, we generated single alanine mutants in the HUS-box (*Figure 6D—figure supplements 3* and *4*). One mutation, N653A, was both inviable in the sensitized strain and compatible with purification. The N653A Sec7ΔC construct showed no difference in GEF activity compared to the wild-type allele in the presence of membranes, indicating that the HUS-box plays no role in the observed exchange rate enhancement by the DCB/HUS domain.

Another potential role of the DCB/HUS domain which would not have been uncovered in our in vitro assay is its interaction with Arl1 (*Christis and Munro, 2012*). Despite extensive effort, we could not detect any stable interaction between *S. cerevisiae* Sec7ΔC and Arl1 in the presence or absence of membranes, nor did preincubation of Sec7ΔC with membrane-bound Arl1 show any effect on catalytic rate (*Figure 4—figure supplements 3–5*). As we previously reported an interaction between Arl1 and longer Sec7 constructs (*McDonold and Fromme, 2014*), this interaction appears to depend on one or more of the HDS domains in addition to the DCB/HUS domain in *S. cerevisiae* Sec7.

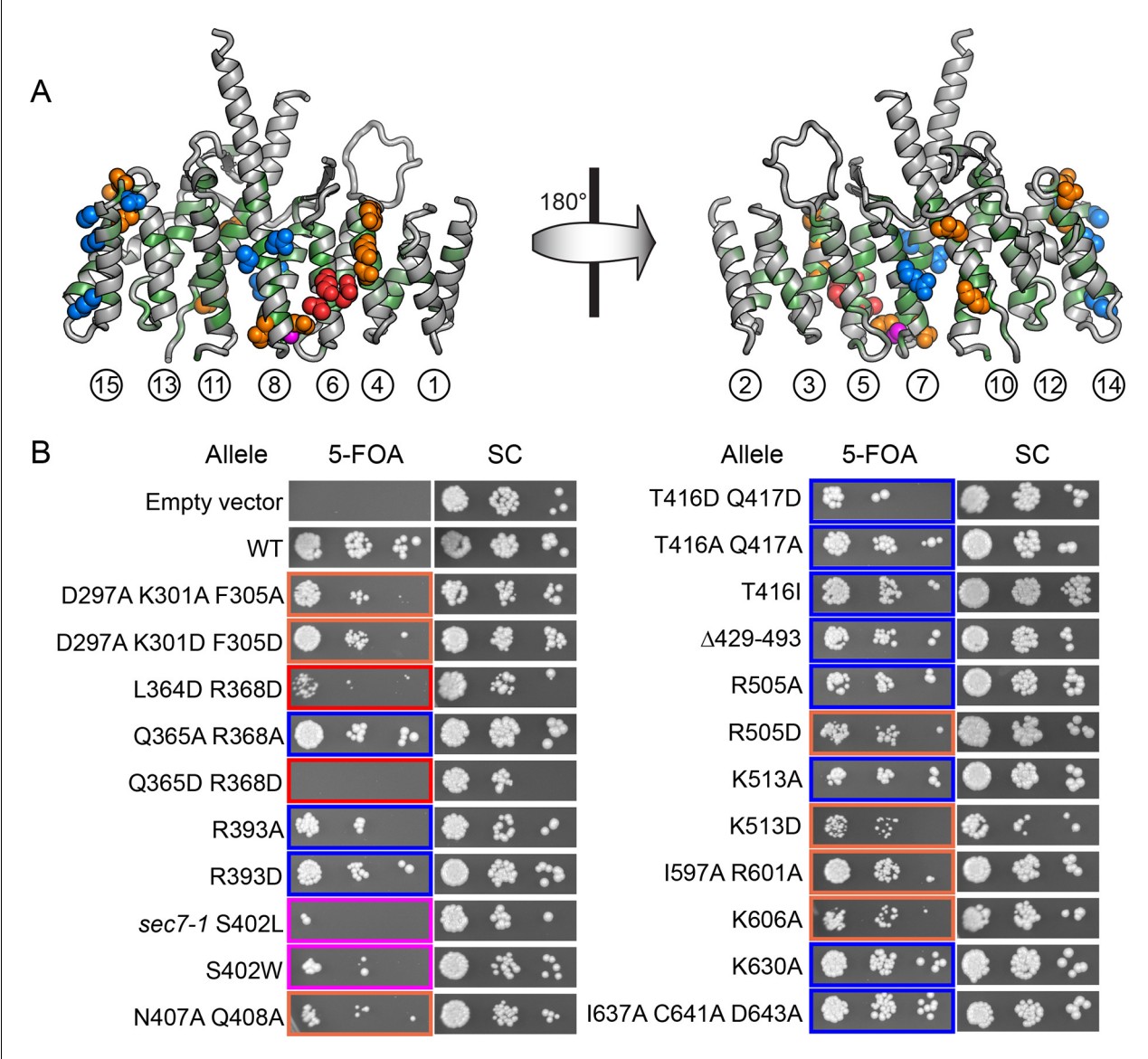

**Figure 5.** Conserved DCB/HUS surface regions mediate Sec7 function. (**A**) Locations of all *S. cerevisiae* mutants tested are shown as space-filling spheres mapped on the *T. terrestris* structure, with backbone colored by conservation. Positions of mutations resulting in stronger temperature sensitive growth phenotypes are colored red (e.g., *S. cerevisiae* Q365/R368), positions with weaker phenotypes are colored orange (e.g., *S. cerevisiae* D297/K301/F305), and positions with no growth phenotype are colored blue; the position corresponding to the *sec7-1* mutation (S402L) is colored magenta. (**B**) Using a plasmid-shuffling assay, CEN plasmids bearing GFP-tagged Sec7 or Sec7$_f$ with the indicated mutations expressed via their endogenous promoter were tested for their ability to rescue a genomic *sec7* deletion in a sensitized *arf1Δ*/*ARF2* strain (CFY863). Growth of serial 10-fold dilutions after 3 days at 37°C is shown, comparing the shuffled strains on 5-FOA to their parents growing in parallel on synthetic complete media (SC). Changes in the number or size of colonies indicates a growth defect.

## The DCB/HUS domain facilitates lipid insertion of the Arf1 amphipathic helix

Finally, to distinguish between different models for lipid-dependent DCB/HUS function, we assayed the ability of the isolated DCB/HUS domain to compete with the Sec7ΔC construct in the GEF activity assay. Addition of a 16-fold excess of DCB/HUS to an exchange reaction inhibited activity to near the levels of the isolated GEF domain, indicative of a function other than direct stimulation of the GEF domain (*Figure 7A*).

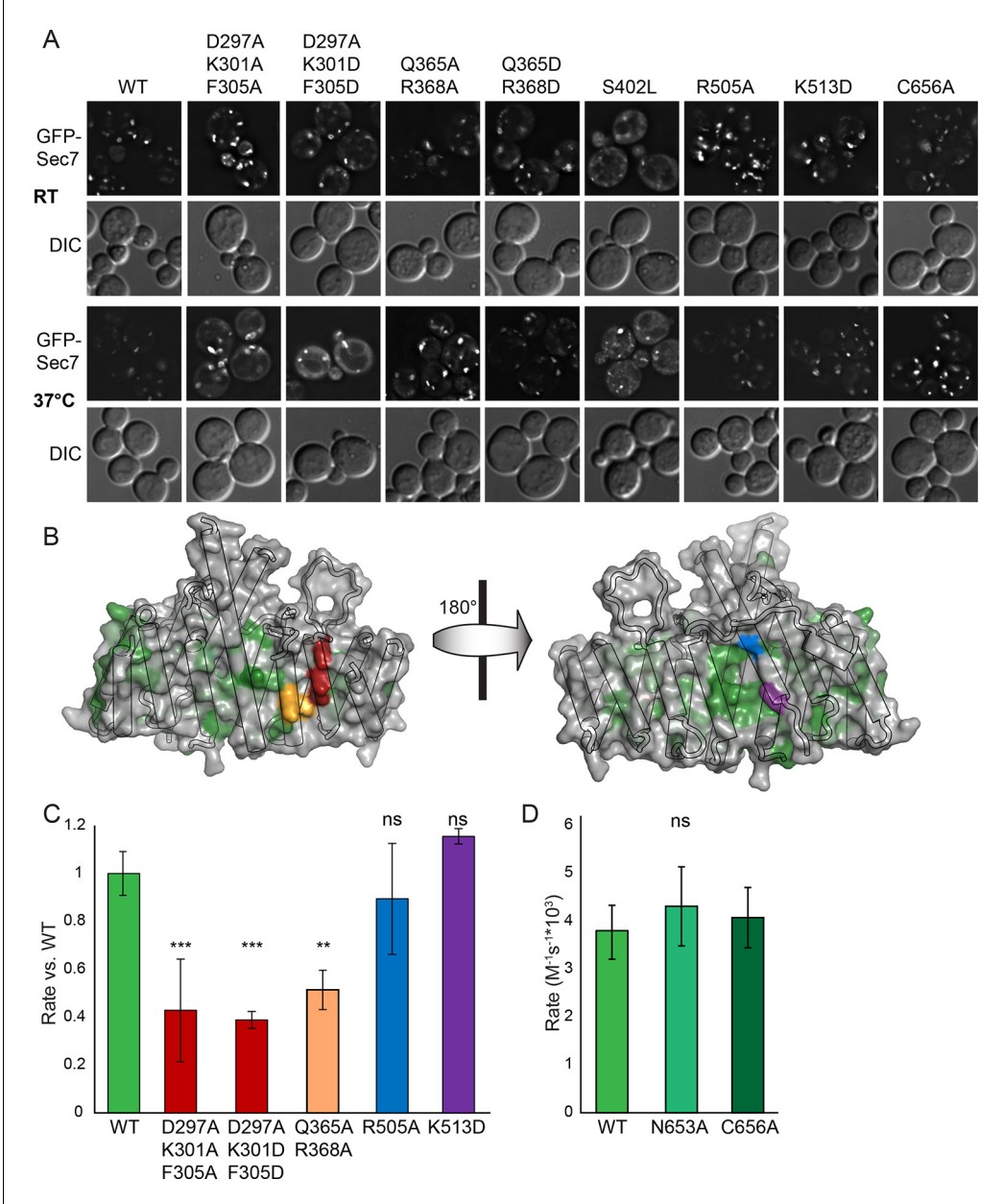

**Figure 6.** Helices 4 and 6 of the DCB/HUS domain mediate GEF stimulation. (**A**) *sec7Δ/arf1Δ* strains bearing the indicated GFP-Sec7 alleles on a centromeric plasmid expressed from the *SEC7* promoter were imaged at permissive and restrictive temperatures. (**B**) Surface residue conservation is shown on the basis of a 361-sequence MUSCLE alignment comprising all BLAST hits of the *T. terrestris* Sec7 N-terminus following removal of incomplete sequences and sequences with more than 95% pairwise identity. Green represents conservation. Residues mutated for biochemical assays are shown in colors matching the resultant bars. A closer view of the residue mutated in *sec7-1* is shown in supplement 1. (**C**) Mutants purifiable as *S. cerevisiae* Sec7ΔC constructs were assayed for rate of nucleotide exchange of myristoylated Arf1 in the presence of liposomes. Activity of the same mutants toward ΔN17Arf1 in the absence of liposomes is shown in supplement 2. (**D**) The two HUS-box mutants purifiable as *S. cerevisiae* Sec7ΔC constructs were assayed for rate of nucleotide exchange of myristoylated Arf1 in the presence of liposomes. Viability and in vivo stability are assessed in supplements 3 and 4.

The following figure supplements are available for figure 6:

**Figure supplement 1.** Atomic basis of the *sec7-1* phenotype.

*Figure 6 continued on next page*

Introduction of mutations into the competing DCB/HUS construct partially alleviated competition correspondent to their inhibitory effects when assayed in the Sec7ΔC construct, confirming the identification of the site responsible for activity (*Figure 7B*). We considered the possibility that the excess competing DCB/HUS domain was simply saturating the membrane surface and therefore interfering with membrane-proximal Arf activation. Therefore, we tested whether addition of excess liposome membranes could relieve the competitive inhibition. Addition of a sixfold excess of membranes failed to relieve even partially the inhibitory effect of excess competing DCB/HUS domain (*Figure 7A*). This indicates that inhibition of Sec7ΔC GEF activity by excess DCB/HUS domain does not occur through competition for binding to the membrane surface. Therefore, excess DCB/HUS domain likely competes with Sec7ΔC for binding to the substrate Arf1.

Taken together, our results are consistent with a direct, and likely transient, interaction of the DCB/HUS domain with the Arf1 substrate and imply that the GEF domain and DCB/HUS domain each make distinct contacts with Arf1 during the exchange reaction. Analysis of SAXS data collected on *S. cerevisiae* Sec7ΔC suggests flexible linkage between the DCB/HUS and GEF domains (*Figure 2—figure supplement 1*), supporting the possibility of simultaneous DCB/HUS and GEF domain binding to Arf1.

## Discussion

In this study, we determined that the regulatory DCB and HUS domains of Sec7 form an unexpected single structural unit that plays a direct role in Arf1 activation at the membrane surface. Previous studies described roles of the DCB/HUS domain in regulation via oligomerization and Arl1 binding (*Christis and Munro, 2012*; *Ramaen et al., 2007*). The strength of the DCB/HUS Arl1 interaction appears to differ among Sec7 orthologs from different species; although Arl1 is clearly important for BIG1/2 localization in fly and human cells (*Christis and Munro, 2012*), it is dispensable for Sec7 localization in *S. cerevisiae* (*McDonald and Fromme, 2014*; *Setty et al., 2004*) and is not required for Sec7 function in all fly tissues (*Torres et al., 2014*). This suggests eukaryotic cells have evolved multiple ways to regulate Arf activation at the TGN using the same Arf-GEF.

We have now described an additional and likely conserved function of the DCB/HUS domain: residues of helices 4 and 6 stimulate Arf1 nucleotide exchange activity dependent on the presence of lipids and the GEF domain. Based on the data we present here, a revised model for DCB/HUS function must account for the requirement of an intact Arf1 membrane insertion helix, for the requirement of lipids, and for a presumably transient interaction with Arf1. One speculative model consistent with all observations is presented in *Figure 7C*. A consequence of Arf1 acting to protect its own hydrophobic membrane insertion moiety tightly coupled with nucleotide state is that the nucleotide exchange transition state may briefly expose the Arf1 N-terminal myristoylated amphipathic helix to aqueous solution before its insertion into the membrane. Exchange of nucleotide by the Sec7 GEF domain may position the flexibly linked DCB/HUS domain appropriately for stabilization of this transition state, either via direct transient binding to the amphipathic helix or via positioning Arf1 optimally relative to the adjacent membrane surface.

The Golgi Arf-GEFs regulate the central membrane trafficking pathway involved in growth of eukaryotic cells by delivering the bulk of lipids to the plasma membrane (*Drubin and Nelson, 1996*). We have previously observed a maximum in vitro Sec7-mediated Arf1 exchange rate of $2 \times 10^6$ M$^{-1}$s$^{-1}$, requiring intact Sec7$_f$ and activating small GTPases (*McDonald and Fromme, 2014*). Using existing estimates of membrane flux and cellular and vesicular protein concentrations (*Chong et al., 2015*; *Dodonova et al., 2015*; *Layton et al., 2011*; *Presley et al., 2002*), we estimate the rate of Arf1 nucleotide exchange required to support maximal cell growth to be approximately $6 \times 10^4$ M$^{-1}$s$^{-1}$ (*Supplementary file 3*). Sec7 must additionally process enough Arf1 to counterbalance endolysosomal trafficking, as well as 'overhead' for non-productive exchanges not leading to vesiculation. While these latter rates are much more difficult to estimate, it is entirely plausible that the four- to eightfold increase in activity provided by the DCB/HUS domain would prove essential to achieving the Arf1 activation throughput required to support maximal cell growth.

While multimerization via the DCB and HUS domains has been observed in several species and has been proposed to regulate Sec7 activity (*Grebe et al., 2000*; *Ramaen et al., 2007*), the ability of the DCB/HUS domain to function as a monomer in the *T. terrestris* Sec7ΔC construct suggests that dimerization is not essential for DCB/HUS domain function. Sec7 is autoinhibited by both its

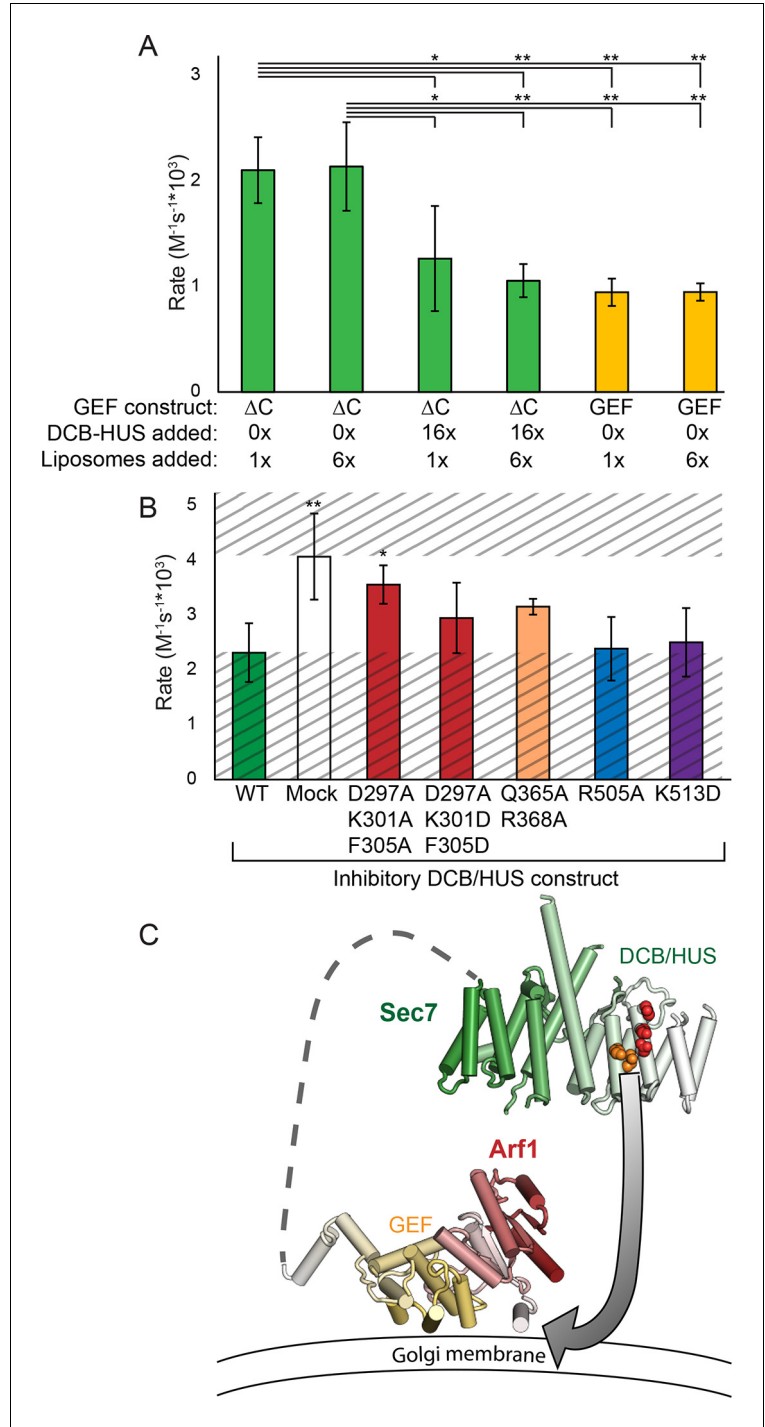

**Figure 7.** The DCB/HUS domain can inhibit GEF activity *in trans.* (**A**) *S. cerevisiae* Sec7ΔC and isolated GEF constructs were assayed for rate of nucleotide exchange of myristoylated Arf1 in the presence of liposomes, with 16-fold excess DCB/HUS construct or sixfold additional liposomes added as indicated. (**B**) Wild-type *S. cerevisiae* Sec7ΔC was assayed for nucleotide exchange of myristoylated Arf1 in the presence of liposomes and a 12-fold excess of DCB/HUS constructs bearing the indicated mutations. The range of activity of interest is bounded by the WT and mock rates and is left unshaded. (**C**) A speculative model of DCB/HUS domain and GEF domain cooperation in Arf1 activation.

HDS1 and HDS4 domains, and appears to adopt open and closed conformations (*McDonold and Fromme, 2014*; *Richardson et al., 2012*). Interactions between the DCB/HUS and HDS domains may provide a mechanism for autoinhibition by occlusion of the GEF domain in a closed conformation. The absence of the HDS domains in a truncated construct would expose an unstable interaction region in the DCB/HUS domain that becomes stabilized by dimerization. Dimerization of the intact DCB/HUS domain may be an artifact of truncation, but perhaps also corresponds to a *bona fide* regulatory conformation of the full-length protein induced through interactions with known regulators (*McDonold and Fromme, 2014*). Our observations that the HDS4 domain mediates both dimerization and autoinhibition imply that Sec7 dimerization is indeed a means of regulation. In the absence of separation of function mutants, it is currently not possible to determine whether dimerization or autoinhibition is the more important function of the HDS4 domain, or whether these two functions are intimately coupled and therefore inseparable. Interestingly, a recent study reported that dimerization of human GBF1, which lacks an HDS4 domain, is not required for its function in cultured cells (*Bhatt et al., 2015*).

We have demonstrated that the DCB/HUS domain of Sec7 stimulates GEF activity by facilitating Arf1 amphipathic helix insertion into a lipid environment during nucleotide exchange. Further studies are now needed to elucidate the role of the HUS-box and how the DCB/HUS domain functions in concert with the HDS domains, as well as the basis for the apparent differences in regulation between the early-Golgi and late-Golgi Arf-GEFs.

## Materials and methods

### Purification of constructs

Sec7 and Arf1 constructs were purified as previously described (*Richardson et al., 2012*; *Richardson and Fromme, 2015*), with no adjustments to the *S. cerevisiae* construct protocols required for *T. terrestris*. Briefly, all *S. cerevisiae* and *T. terrestris* Sec7 constructs were expressed with an N-terminal $His_6$ tag in 2–12 L cultures of *E. coli* in Terrific Broth, grown overnight at 18°C. Following lysis, proteins were purified via Ni.NTA resin (Qiagen) in batch, followed by MonoQ ion exchange and Superdex 200 gel filtration (GE Healthcare), with a final buffer composition of 20 mM HEPES pH 7.5, 150 mM NaCl, and 2 mM DTT.

ΔN17Arf1 constructs were purified as per Sec7, with additional 2 mM $MgCl_2$ added to all buffers. Full-length Arf1 was coexpressed with NMT1 to myristoylate, then following lysis was purified via batch incubation with DEAE-sephacel, batch incubation with ToyoPearl phenyl resin, and Superdex 200 gel filtration. Arf1 L8K purification followed an almost identical protocol, but without NMT1 coexpression. Myristoylated Arf1 was assumed to be fully GDP-bound; Arf1 L8K and ΔN17Arf1 were treated with EDTA in the presence of excess GDP to convert them to their GDP-bound form prior to activity assays.

### Crystallization and structure determination

Selenomethionine substituted *T. terrestris* $His_6$-Sec7(1–458) at 16 mg/ml was crystallized via the hanging drop method, mixing 1:1 with 6% Jeffamine ED-2001, 150 mM MES pH 6, 3% DMSO, and 10 mM DTT at 4°C. Crystals were cryoprotected in a three-stage shift to synthetic well solution plus 30% DMSO. The native construct was crystallized and cryoprotected similarly, using a well solution of 5% Jeffamine ED-2001, 150 mM MES pH 6, and 6% DMSO.

Diffraction data were collected locally at CHESS beamline A1 on an ADSC Quantum-210 CCD detector and processed using HKL-2000 (*Otwinowski and Minor, 1997*); to improve completeness of collected reflections, datasets from two crystals from the same drop were merged to obtain the final native dataset. Experimental phases were obtained via single-wavelength anomalous diffraction using PHENIX (*Adams et al., 2010*), permitting the auto-building of the alpha-helical structure by PHENIX, followed by serial manual building in Coot (*Emsley et al., 2010*) and re-refinement including TLS modeling (*Painter and Merritt, 2006*). The anomalous signal permitted assignment of residues 245–271 of the interdomain loop, containing two asymmetrically positioned methionines. All protein models were visualized using PyMol (Schrödinger).

## Small-angle X-ray scattering (SAXS)

Small-angle X-ray scattering data were collected at CHESS beamline F2 at room temperature on samples purified to homogeneity as described. Multiple serial exposures of each construct at 4 mg/ml and 3–4 successive twofold dilutions were collected against exactly matched blanks and processed using BioXTAS RAW (*Nielsen et al., 2012*) to confirm absence of radiation- or concentration-dependent aggregation. Theoretical models based on crystal structures were calculated and fit to the experimental data using BUNCH and CORAL (*Petoukhov et al., 2012*; *Petoukhov and Svergun, 2005*)

## Multi-angle light scattering (MALS)

Proteins purified to homogeneity were exchanged into fresh buffer by serial concentration and redilution to a final concentration of 5 mg/ml, and run through a Wyatt WTC-050S5 gel filtration column coupled to DAWN HELEOS-II light scattering and Optilab T-rEX refractive index detectors (Wyatt Technology) at room temperature. Data were analyzed via ASTRA 6 software to obtain the molecular weight of the sample, and compared to that predicted from the sequence to determine oligomeric state.

## Arf1 nucleotide exchange kinetics

Arf1 nucleotide bound state was measured in real-time as described previously (*Richardson et al., 2012*), with the exception that [Sec7] was increased to equimolar with [Arf1] to permit accurate measurement of the slow GEF construct reaction rate and avoid potentially confounding results from unstable mutant constructs. To a final reaction volume of 150 µl, HKM buffer (20 mM HEPES pH 7.5, 150 mM KOAc, 2 mM $MgCl_2$), 200 µM lipids (lipsomes or micelles), 670 nM Sec7 construct, 670 nM Arf1 construct, and 200 µM GMPPNP were sequentially added while monitoring fluorescence, waiting 1–3 min between steps for the fluorescence to stabilize. After addition of GMPPNP, the fluorescence was monitored for an additional 40 min and fit to a single exponential curve to determine the rate of exchange. In the absence of Sec7, fluorescence traces were observed to remain stable, indicating that the observed changes in fluorescence were solely due to the activity of Sec7, and were not a lipid-mediated effect. The calculated rate of exchange was then normalized to the native fluorescence of the input Sec7 construct to obtain $k_{react}$. Note that the per-molecule exchange rates calculated from these reactions are lower than reported previously, in part due to the higher concentrations of GEF constructs used here and variations in behavior between liposome batches. Triplicates of each condition were collected for statistics; error bars represent 95% confidence as determined by t test or ANOVA with Tukey's or Dunnett's test in postprocessing, as appropriate.

## Yeast plasmid shuffle

Sec7 alleles to be assayed were cloned with an N-terminal GFP tag into pRS415 (*LEU2* marked), and transformed into yeast containing genomic *sec7* and *arf1* disruptions and a *URA3* plasmid harboring wild-type *SEC7* (CFY863). Following overnight growth in –Leu synthetic media, serial half-log dilutions were spotted on 5-FOA and synthetic complete or dropout media and grown for three days at 37°C.

## Microscopy

Cells were grown in synthetic dropout media and imaged in log phase ($OD_{600} \sim 0.5$). For the temperature shift experiment, cells were grown shaking in a 37°C water bath for 30 min before imaging. Live cells were imaged on a DeltaVision RT wide-field deconvolution microscope (Applied Precision). Images were deconvolved using SoftWoRx 3.5.0 software (Applied Precision). Images were further processed in ImageJ, adjusting only min/max light levels for clarity, and using equivalent processing for all images within an experiment.

## Liposome pelleting and flotation experiments

Membrane binding assayed by liposome pelleting and flotation experiments was performed as described (*Richardson et al., 2012*; *Paczkowski et al., 2012*; *Richardson and Fromme, 2015*). Additional details are provided in the corresponding figure legends.

## Accession number

The PDB accession number for the *T. terrestris* Sec7 DCB-HUS domain is 5HAS.

## Acknowledgements

We thank members of the Sondermann lab and R Gillilan and the A1 beamline staff at CHESS for assistance with data collection. We also thank members of the Emr and Fromme labs for helpful discussions, and S Tang for help with the manuscript. CHESS is supported by the NSF & NIH/NIGMS via NSF award DMR-1332208, and the MacCHESS resource is supported by NIH/NIGMS award GM103485. The authors were supported by NIH award R01GM098621.

## Additional information

### Funding

| Funder | Grant reference number | Author |
| --- | --- | --- |
| National Institutes of Health | R01GM098621 | Brian C Richardson<br>Steve L Halaby<br>Margaret A Gustafson<br>J Christopher Fromme |
| National Institutes of Health | T32GM007273 | Steve L Halaby<br>Margaret A Gustafson |

The funders had no role in study design, data collection and interpretation, or the decision to submit the work for publication.

### Author contributions

BCR, JCF, Conception and design, Acquisition of data, Analysis and interpretation of data, Drafting or revising the article, Contributed unpublished essential data or reagents; SLH, Acquisition of data, Analysis and interpretation of data, Drafting or revising the article; MAG, Drafting or revising the article, Contributed unpublished essential data or reagents

### Author ORCIDs

Brian C Richardson, http://orcid.org/0000-0002-5434-1500
J Christopher Fromme, http://orcid.org/0000-0002-8837-0473

## Additional files

### Supplementary files

• Supplementary file 1. *T. terrestris* intron assignment *C. thermophilum*, *M. thermophila*, and *T. terrestris* Sec7 genomic sequences, each containing a single annotated intron in the Sec7ΔC region, are aligned with annotated introns shown in lowercase. Conservation suggests that the *T. terrestris* intron should instead match that of the other two species; the intron assignment assumed for this work is highlighted in gray, and this correction to the construct was required for its expression (not shown).

• Supplementary file 2. Plasmids and strain tables

• Supplementary file 3. Estimation of exocytic Arf1 flux. Values for Arf1 trafficking calculations are provided.

### Major datasets

The following datasets were generated:

| Author(s) | Year | Dataset title | Dataset URL | Database, license, and accessibility information |
|---|---|---|---|---|
| Richardson BC, Fromme JC | 2015 | Crystal structure of the N-terminal DCB-HUS domain of T. terrestris Sec7 | http://www.rcsb.org/pdb/search/structid-Search.do?structureId=5HAS | Publicly available at RCSB Protein Data Bank ( Accession no: 5HAS). |

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
