## [Decision Letter]

Thank you for submitting your work entitled "The Sec7 N-terminal regulatory domains facilitate membrane-proximal activation of the Arf1 GTPase" for consideration by *eLife*. Your article has been reviewed by three peer reviewers, one of whom is a member of our Board of Reviewing Editors. The evaluation has been overseen by the Reviewing Editor and Randy Schekman as the Senior Editor.

The reviewers have discussed the reviews with one another and the Reviewing editor has drafted this decision to help you prepare a revised submission.

Summary:

A structure of the *T. terrestris* DCB/HUS domain of Sec7 is described and the role of these domains in GEF activity toward Arf1 has been investigated. The structure work is well done and is clearly presented. in vitro GEF activity assays show that nucleotide exchange on Arf1 by the Sec7 GEF domain is potentiated by the adjacent DCB/HUS domain in the presence of liposomes, and that this requires the N-terminal helix of Arf1. The authors conclude that the DCB/HUS domain facilitates lipid insertion of the Arf1 amphipathic helix.

Essential revisions:

The reviewers agreed that the major conclusion of the paper, that the DCB/HUS domain facilitates lipid insertion of the Arf1 amphipathic helix, is not adequately supported by the data. While they consider the authors' interpretation of their data to be reasonable, it is not directly supported with experimental data; all that is shown is Trp fluorescence measurements as a proxy for nucleotide exchange. This is necessary to address, particularly since early work on Arf1 (see below) raises an alternative interpretation than the authors' whereby the DCB/HUS domain increase the specific activity of the GEF domain without directly promoting membrane binding by the N-terminal helix. Related, the reviewers are also concerned that the Trp fluorescence assay used to infer nucleotide exchange may be misleading when assayed in the presence of lipids due to the potential that these agents influence the local environment of the Trp residue that reports the occupancy state of the nucleotide binding site. In order to be further considered by *eLife*, the paper will need to provide direct evidence supporting the model that the DCB-HUS domain acts directly on the N-terminal helix of Arf to promote its insertion into the membrane. With regard to the effects of lipids on Arf Trp fluorescence, the reviewers noted that this latter concern might be adequately addressed by a more thorough description of the assay methods, including the strategy used to account for changes in fluorescence not due to nucleotide binding (i.e., in the text).

Background: DPMC micelles stimulate the *intrinsic* rate of hydrolysis on Arf1 (Khan and Gilman 1986). This underscores a concern with the authors' models. Does the DCB-HUS domain promote Arf1 membrane insertion or, alternatively, does DCB-HUS increase the specific activity of the GEF domain when presented when the N-domain of Arf1 is – through whatever mechanism – already pried away from its pocket in Arf1:GDP. Indeed, that specific conformational change (characterized by NMR) has been suggested to be a prerequisite for GTP binding:

"The [yeast Myr-Arf1:GDP] structure clearly places the myristoyl chain in a hydrophobic groove between a C-terminal helix and a loop connecting b strands 3 and 4. The positioning suggests a clash with this loop would occur as it exists in the GTP-bound form, a fact that may require membrane association to accommodate an expelled myristoyl chain before GDP exchange can occur." (Liu, Khan, Prestegard PMID: 19141284). A possibility consistent with the authors' membrane-first model is that the DCB-HUS domain might actually allow Arf1-GDP to stably associate with the membrane (in the absence or presence of the GEF domain). That might be a pretty straightforward floatation or pelleting experiment.

---

## [Author Response]

We thank the editor and reviewers for their supportive and constructive comments, which we have addressed through additional experiments and changes to the text. In our view the primary claim of this study, namely that the DCB/HUS domain of Sec7 forms a single structural unit that facilitates activation of Arf1 at the surface of a membrane, is now firmly supported by the data.

We have performed additional experiments to address the core issue raised by the reviewers, namely to provide direct evidence (aside from Trp fluorescence) that the DCB/HUS domain facilitates Arf1 membrane association (and not just nucleotide exchange). We now show that this is indeed the case, using a membrane-binding (flotation) experiment. We also explain how the fluorescence experiments are controlled to avoid potential confounding artifacts arising from lipids.

*The reviewers agreed that the major conclusion of the paper, that the DCB/HUS domain facilitates lipid insertion of the Arf1 amphipathic helix, is not adequately supported by the data. While they consider the authors' interpretation of their data to be reasonable, it is not directly supported with experimental data; all that is shown is Trp fluorescence measurements as a proxy for nucleotide exchange. This is necessary to address, particularly since early work on Arf1 (see below) raises an alternative interpretation than the authors' whereby the DCB/HUS domain increase the specific activity of the GEF domain without directly promoting membrane binding by the N-terminal helix. Related, the reviewers are also concerned that the Trp fluorescence assay used to infer nucleotide exchange may be misleading when assayed in the presence of lipids due to the potential that these agents influence the local environment of the Trp residue that reports the occupancy state of the nucleotide binding site. In order to be further considered by* eLife*, the paper will need to provide direct evidence supporting the model that the DCB-HUS domain acts directly on the N-terminal helix of Arf to promote its insertion into the membrane. With regard to the effects of lipids on Arf Trp fluorescence, the reviewers noted that this latter concern might be adequately addressed by a more thorough description of the assay methods, including the strategy used to account for changes in fluorescence not due to nucleotide binding (i.e., in the text).*

*Background: DPMC micelles stimulate the* intrinsic *rate of hydrolysis on Arf1 (Khan and Gilman 1986). This underscores a concern with the authors' models. Does the DCB-HUS domain promote Arf1 membrane insertion or, alternatively, does DCB-HUS increase the specific activity of the GEF domain when presented when the N-domain of Arf1 is* – *through whatever mechanism – already pried away from its pocket in Arf1:GDP. Indeed, that specific conformational change (characterized by NMR) has been suggested to be a prerequisite for GTP binding:*

*"The [yeast Myr-Arf1:GDP] structure clearly places the myristoyl chain in a hydrophobic groove between a C-terminal helix and a loop connecting b strands 3 and 4. The positioning suggests a clash with this loop would occur as it exists in the GTP-bound form, a fact that may require membrane association to accommodate an expelled myristoyl chain before GDP exchange can occur." (Liu, Khan, Prestegard PMID: 19141284). A possibility consistent with the authors' membrane-first model is that the DCB-HUS domain might actually allow Arf1-GDP to stably associate with the membrane (in the absence or presence of the GEF domain). That might be a pretty straightforward floatation or pelleting experiment.*

Regarding possible artifacts arising from any affects of lipids on Arf Trp fluorescence, we can rule these out with certainty based on control experiments. To convince the reader of this, we now include additional descriptions in the Results and Methods sections. The most important thing to note is that we do directly monitor how the addition of each component changes the fluorescence of the reaction. Notably, exchange rates are extracted from time-course changes in kinetics that only occur subsequent to the addition of GTP, and we include a baseline control (i.e., no GEF construct) in each set of experiments to confirm that no change in fluorescence is seen over time in the absence of exchange (see Figure 4, black trace). Any contributions to Trp fluorescence from lipids themselves will impact the magnitude of the fluorescence readings, but not the rate at which fluorescence changes over time. The rate of change in fluorescence over time is a direct read-out of Arf conformational change associated with binding to GTP.

We now provide additional experimental evidence for the role of DCB/HUS in promoting Arf1 association with membranes (which occurs through insertion of the N-terminal amphipathic helix). In the new Figure 4—figure supplement 1, we show a membrane-association time-course, as assayed by liposome flotation, comparing the ability of Sec7 GEF constructs to stimulate stable Arf1 membrane binding (in the presence of GTP). The Sec7ΔC construct (containing the DCB/HUS domain) is much more active than the isolated GEF domain in this assay. The extent of the membrane- binding directly parallels the extent of Arf activation (as monitored by TRP fluorescence) for both GEF constructs, demonstrating that the DCB/HUS domain facilitates both Arf1 activation and stable membrane association (which requires insertion of the N-terminal amphipathic helix). We also performed the control experiments suggested by Reviewer 3 to provide additional evidence in support of the model (see below).

We note that our experiments do directly address the possibility that the DCB/HUS domain might simply increase the specific activity of the GEF domain: The fact that the DCB/HUS domain does not stimulate the activity of the GEF domain towards the deltaN17-Arf1 or L8K-Arf1 substrates (Figure 4) demonstrates that the DCB-HUS domain is acting only when the N-terminus of Arf1 needs to be inserted into lipids. Furthermore, as the deltaN17-Arf1 substrate lacks the N-terminal helix of Arf1, this helix can be considered to be “pried away” from its binding pocket in this mutant substrate, yet the DCB/HUS domain has no effect on this substrate.

Additionally, the competition experiments presented in Figure 7 demonstrate that excess DCB/HUS domain inhibits the activity of the Sec7- deltaC construct, again arguing against a simple increase in the specific activity of the GEF domain.

We agree that it is very likely that membrane association of Arf1- GDP occurs prior to the exchange reaction. However, this interaction is known to be very transient (Antonny et al., Biochemistry 1997). If the DCB/HUS domain stabilizes this transient intermediate state (which we think it does), this state is still likely to be quite transient compared to the time-scale of a pelleting or flotation experiment. We previously performed an experiment similar to that suggested above – as reported in our 2012 paper (Figure 2), Arf1-GDP does not bind stably to membranes in the presence of any Sec7 constructs, including Sec7-deltaC. Our GEF assays are perhaps more informative in this regard, as an increased rate of reaction is suggestive of the stabilization (lower energy) of a high-energy intermediate or transition state.

We note that we propose two alternatives for how the DCB/HUS domain might facilitate Arf1 membrane-proximal activation and insertion of the amphipathic helix: either through direct transient interaction with the Arf1 N-terminus or by positioning Arf1 optimally for membrane insertion (either binding to a different region of Arf1 or otherwise constraining the orientation of the GEF domain in a manner that would position the Arf1 N- terminal helix close to the membrane when Arf1 is engaged by the GEF domain). We think that both of these possibilities are interesting, as either involves function of the DCB/HUS domain to overcome the activation barrier associated with membrane insertion.

We expect that further determination of exactly how the DCB/HUS domain facilitates membrane insertion of the Arf1 N-terminus will await structures of longer constructs of Sec7 or its homologs “caught in the act” of activating Arf1.